# Alternative Strategies to Inhibit Tumor Vascularization

**DOI:** 10.3390/ijms20246180

**Published:** 2019-12-07

**Authors:** Alessia Brossa, Lola Buono, Sofia Fallo, Alessandra Fiorio Pla, Luca Munaron, Benedetta Bussolati

**Affiliations:** 1Department of Molecular Biotechnology and Health Sciences, Universitty of Torino, 10126 Torino, Italy; alessia.brossa@unito.it (A.B.); lola.buono@unito.it (L.B.); sofia.fallo@unito.it (S.F.); 2Department of Life Science and Systems Biology, University of Torino, 10126 Torino, Italy; alessandra.fiorio@unito.it (A.F.P.); luca.munaron@unito.it (L.M.)

**Keywords:** tumor vasculogenesis, endothelial cells, anti-angiogenic drugs, normalization, endothelial demesenchymalization, endothelial vaccination

## Abstract

Endothelial cells present in tumors show different origin, phenotype, and genotype with respect to the normal counterpart. Various mechanisms of intra-tumor vasculogenesis sustain the complexity of tumor vasculature, which can be further modified by signals deriving from the tumor microenvironment. As a result, resistance to anti-VEGF therapy and activation of compensatory pathways remain a challenge in the treatment of cancer patients, revealing the need to explore alternative strategies to the classical anti-angiogenic drugs. In this review, we will describe some alternative strategies to inhibit tumor vascularization, including targeting of antigens and signaling pathways overexpressed by tumor endothelial cells, the development of endothelial vaccinations, and the use of extracellular vesicles. In addition, anti-angiogenic drugs with normalizing effects on tumor vessels will be discussed. Finally, we will present the concept of endothelial demesenchymalization as an alternative approach to restore normal endothelial cell phenotype.

## 1. Introduction

### 1.1. Tumor Endothelial Cell Characterization

In 1971, Judah Folkman observed that solid tumors show a diffuse vascular network, often hemorrhagic, and that poorly vascularized tumors were unable to grow beyond 2–3 mm [1]. These observations led Folkman to hypothesize that to grow and expand, tumors need new blood vessels, introducing the concept of tumor angiogenesis [1]. It is now well established that, during the early phases of tumor angiogenesis, a process called “angiogenic switch” occurs, characterized by overexpression of pro-angiogenic factors, neoangiogenesis, and tumor cell survival [2]. Indeed, tumor angiogenesis significantly differs from physiological angiogenesis. Tumor vessels have an irregular aspect, are dilated and tortuous, and this chaotic organization results in the absence of distinct venules, arterioles, and capillaries, with the formation of a leaky and hemorrhagic vascular network [3]. They have an incomplete basal membrane, with large joints and fenestrations which increase the interstitial fluid pressure, possibly resulting in intra-tumor bleeding [4].

Tumor endothelial cells (TEC) themselves substantially differ from the normal counterpart. In 2000, Croix et al. identified for the first time genes differentially expressed in TEC with respect to normal endothelial cells, most of which are involved in the formation of collagen, in angiogenesis and in the wound healing process, demonstrating that tumor endothelium is different from normal endothelium at a molecular level [5]. In addition, TEC can be aneuploid, express embryonic markers, and can undergo endothelial–mesenchymal transition. Hida et al. first demonstrated that freshly isolated TEC present structural aberrations, such as nonreciprocal translocations, missing chromosomes, and have multiple centrosomes [6]. Functionally, TEC display an increased proliferation rate and delayed senescence with respect to normal endothelial cells due to autocrine production of proangiogenic factors [7] and are resistant to classical anti-angiogenic drugs [8,9].

### 1.2. Tumor Endothelial Cell Origin

The evidence that tumor vessels differ from normal vessels, both, genotypically and functionally suggests that tumor vasculature could either be modified by factors deriving from the tumor microenvironment or directly originate by intra-tumor vasculogenesis as alternative mechanisms other than the recruitment from pre-existing vessels in adjacent tissues [10]. The strategies of intra-tumor vasculogenesis are shown in Figure 1.

There are several pieces of evidence that factors secreted by tumor cells, and *in primis* extracellular vesicles (EVs), may reprogram normal quiescent endothelial cells through the transfer of proteins and genetic material (mRNAs, miRNAs, or proteins) [11,12,13]. In parallel, the intratumor vasculogenesis might be dependent on the differentiation of normal or cancer stem cells or by endothelial mimicry of differentiated tumor cells [10]. Bone marrow-derived cells, and in particular endothelial progenitor cells, actively participate to tumor growth, not only through the secretion of pro-angiogenic factors but also through their incorporation within the vessels [14,15]. Resident normal tissue stem cells were also shown to differentiate into endothelial cells in the presence of growth factors released by the tumor [15]. Cancer stem cells (CSC), a subpopulation of tumor cells with stem properties, can generate all different tumor cell types, becoming responsible for tumor growth and progression. Several groups demonstrated the ability of CSC to differentiate into endothelial cells and pericytes and thus their contribution to tumor vasculogenesis [16,17,18].

Differentiated cancer cells themselves can also generate vascular structures by a process called vasculogenic mimicry. First identified in melanoma [19], the presence of vascular mimicry has been subsequently confirmed in a number of tumors, such as lung, breast, prostate, bladder, and renal carcinomas and glioblastoma [20]. Finally, to rapidly adapt to the surrounding microenvironment, tumors may generate new vessels trough intussusceptive microvascular growth. This mechanism, also known as non-sprouting or splitting angiogenesis, is characterized by the generation of new blood vessels by splitting an existing one [21]. The capillary network can, therefore, increase its complexity and vascular surface, generating vessels more rapidly with a minor metabolic demand as compared to sprouting angiogenesis.

Given the different origin, phenotype, and genotype of TEC with respect to the normal counterpart, in the last decades, many researchers focused on the isolation of TEC from solid tumors (Table 1) [22], to obtain an in vitro model resembling tumor angiogenesis.

### 1.3. Classic Anti-Angiogenic Therapies

A number of anti-angiogenic drugs have been developed and proposed to limit tumor growth and expansion [34]. At present, the main anti-angiogenic therapies approved by the FDA are described in Table 2 [34]. The use of anti-angiogenic drugs in clinical practice, however, only showed an initial benefit in patients, followed by limited effectiveness and only a moderate disease-free survival [35]. This is mainly due to the expression of alternative angiogenic pathways [36,37]. Although inhibitors of the VEGF pathway are substantially effective in reducing tumor vascularization, after treatment discontinuation the tumor vascular network is able to re-grow, acquiring overexpression of vascular growth factor receptors [36]. This overexpression leads the survived vessels to VEGF-independency and, therefore, to the development of resistance [37]. In addition, anti-angiogenic treatment can lead to the formation of a hypoxic microenvironment, which regulates the cancer stem cell population and can contribute both to the maintenance of the tumor and to the resistance to therapies [36].

Anti-angiogenic agents, such as the monoclonal antibody (mAb) bevacizumab, only showed significant activity when combined with cytotoxic chemotherapy [36]. Moreover, despite the success of the dual blockade of VEGFR and PDGFR by the tyrosine kinase inhibitor (TKI) sunitinib, a combination strategy using bevacizumab and imatinib, another inhibitor of PDGF signaling, was not effective but rather toxic during renal cancer treatment [38]. After 10 years of approval by the FDA of the first anti-VEGF drug, bevacizumab, resistance to anti-VEGF therapy remains a challenge in the treatment of cancer patients, revealing the need to explore alternative strategies to classical anti-angiogenic therapies, to obtain a durable therapeutic effect. In this review, we will describe some alternative strategies to inhibit tumor vascularization, such as the use of new mAbs, the target of alternative signaling pathways, the vaccination with endothelial antigens, and the use of extracellular vesicles (Figure 2). In addition, the use of alternative anti-angiogenic drugs with normalizing effects on tumor vessels will be described.

## 2. Alternative Molecular Targets

### 2.1. Alternative Anti-Angiogenic Antibodies

The therapeutic use of classical anti-angiogenic drugs, as the anti-VEGF Ab bevacizumab, lacked the expected results observed in experimental models [35]. As mentioned earlier, anti-VEGF therapy-resistant tumors increase the expression of molecules that activate alternative angiogenic pathways [36,37,39], they can represent a new target of antibody-mediated therapies. For example, Abs against anti-angiopoietin-2 (like nesvacumab, AMG780, MEI3617, and vanucizumab), and Abs anti-integrin αvβ3 have been successfully tested in phase I/II study [40,41]. Another example of an antibody targeting alternative angiogenic pathways is MP0250, a genetically engineered designed ankyrin repeat protein (DARPin^®^) that specifically binds to VEGF-A, hepatocyte growth factor (HGF), and human serum albumin (HSA) [42,43]. The antibody is currently being studied in phase I and II clinical trials on multiple myeloma relapses (NIH N. NCT03136653) [44] on EGFR-mutated non-small cell lung cancer (NIH N. NCT03418532) and on other neoplasms (NIH N. NCT02194426). With its target specificities, MP0250 may thus help to overcome the resistance due to single targeting mAbs. Moreover, transforming growth factor (TGF-β) pathway has been found over-expressed after anti-VEGF therapy [45], suggesting that it might play an important role in the acquisition of therapy resistance. Endoglin (CD105) is a cell membrane glycoprotein overexpressed on proliferating endothelial cells that binds several factors of the TGF-β superfamily, suggesting that activation of this pathway may be responsible for tumor VEGF-independency [46]. In 1995, CD105 was described as a receptor overexpressed in tumor vasculature [47] and, more recently, it has been shown that high CD105 expression on vessels is correlated with poor prognosis in many solid tumors, such as kidney [48], prostate [49], and ovarian cancer [50]. CD105 was also described as a marker of CSC in renal cell carcinoma [16]. TRC105 (carotuximab) is a novel, clinical-stage antibody against CD105, that inhibits tumor vessel formation through the blockade of CD105. In a recent study, the TRC105 effect on both TEC lines and CSC-TEC was described [51]. In particular, TRC105 alone affected the ability of TEC and CSC-TEC to organize in tubular structures [51]. Moreover, TRC105 increased the effect of the tyrosine kinase inhibitor Sunitinib in inhibiting tumor endothelial proliferation, survival, and new vessel formation [51]. Taken together, these findings indicate that the combined inhibition of VEGF and TGF-β pathways may have potential use in renal carcinoma therapy. Indeed, TRC105 is currently being studied in phase III clinical trial in combination with pazopanib for the treatment of advanced angiosarcoma [52] and in multiple phase I and II clinical trials combination with VEGF inhibitors for the treatment of different solid tumors. For example, clinical trials in phase I and II are testing the efficacy of TRC105 in combination with bevacizumab in refractory gestational trophoblastic neoplasia and choriocarcinoma (NIH N. NCT02396511) [53], metastatic renal cancer (NIH N. NCT01727089), and glioblastoma (NIH N. NCT01564914 and NCT01648348). Other studies tested the combination of TRC105 with tyrosine kinase inhibitors, such as Axitinib in renal cell carcinoma (NIH N. NCT01806064), and sorafenib in hepatocellular carcinoma [54]. Encouraging evidence of activity to date was observed, and the study is now continuing to recruit in the phase II stage to confirm the activity of the combination therapy [54]. 

### 2.2. Ca^2+^-Permeable Channels

Accumulating evidence demonstrates that the development of several cancers involves altered Ca^2+^ homeostasis and aberrant ion channel expression [55,56]. This is not surprising considering the multifaceted role of Ca^2+^ as an ubiquitous second messenger, which is involved in the tuning of multiple fundamental cellular functions [57]. Indeed, ion channels represent good potential pharmacological targets due to their location on the plasma membrane, where they can be easily accessed by drugs. As the first reports suggesting a role for ion channels in cancer progression, the field has undergone an exponential development giving rise to a large consensus in the scientific community to include “channelopathy” among the causal factors in cancer development [58,59]. In particular, it has been clearly established a key role for Ca^2+^-permeable channels in tumor vascularization both in vitro and in vivo [60,61,62,63]. Many pro-angiogenic growth factors, as well as chemokines, trigger Ca^2+^ signals directly involved in the angiogenic switch by mediating endothelial cells proliferation, migration, and sprouting [64,65,66,67].

Among Ca^2+^-permeable channels, transient receptor potential (TRP) superfamily has been deeply investigated for their functions in endothelial cells where they emerged as important factors contributing to several key vascular processes, such as vascular tone, permeability, and cell migration [68,69,70]. In addition, different TRP channels have been described as mediators of VEGF-mediated Ca^2+^ signals [71,72]. As previously stated, TEC significantly differ from healthy endothelial cells, showing aberrant phenotypes and physiology. It is, therefore, expected that Ca^2+^ homeostasis is also severely altered in TEC; indeed, Ca^2+^ signals mediated by different growth factors, such as VEGF or ATP and their downstream second messengers (arachidonic acid, nitric oxide, hydrogen sulfide, and cyclic AMP) are drastically remodeled in TEC where they play key roles in cell migration as compared to healthy endothelial cells [73,74,75,76,77]. Intriguingly, recent studies reported that TRP channels are differentially expressed in TEC. In particular, TRPV4 has been shown to exert a proangiogenic role on TEC by promoting cell migration and normalization [78,79]. On the contrary, TRPM8 exerts a protective role in endothelium by inhibiting cell migration via a Rap1/βintegrin mechanism [80]. Comparative TRP expression profile has been recently performed on prostate cancer TEC (PTEC) and their heathy counterpart, as well as on other TEC and endothelial cells. Interestingly, TRPA1, TRPV2, and TRPC3 are overexpressed in PTEC. TRPA1 showed a clear proangiogenic role by promoting an increase in intracellular [Ca^2+^] and consequent endothelial cells migration in vitro as well as sprouting angiogenesis in the retina in vivo model [81].

### 2.3. ERG

ERG (ETS related gene) is part of the E-26 transformation specific (ETS) family of transcription factors, and it was first discovered in 1987 by Reddy et al. in human colorectal carcinoma cells [82]. These factors function as either transcription activators or repressors, depending on the target gene or on the post-transcriptional modification required [83]. From the embryonic developmental stage, ERG is widely expressed in a variety of mesodermic tissues, and in particular in the endothelium, where it‘s highly expressed in the endothelial cells of the majority of adult tissues [83]. In endothelial cells, ERG plays a key role in the regulation of endothelial homeostasis by influencing numerous biological processes, such as vasculogenesis, angiogenesis, junction stability, cell migration, and survival [83]. In fact, ERG has been shown to act as a controller of the balance between pro- and anti-angiogenic processes, by regulating the expression of key genes like VEGFR1, VEGFR2, FZDL4, and EGF-like protein 7 [83]. Furthermore, in vitro studies have shown that ERG is essential for endothelial tube formation [84]. For instance, ERG inhibition studies in human endothelial cells revealed a lowered expression of the adhesion molecule VE-cadherin that resulted in the loss of cell–cell contacts, cell death, and, therefore, malformation of endothelial tubes [84]. These results were confirmed in vivo, whereby a postnatal deletion of ERG in inducible endothelial-specific ERG knockout mice led to defective angiogenesis in the retina, therefore confirming the crucial role of ERG in the regulation of angiogenesis [84,85].

Many studies reported the involvement of abnormal ectopic expression of ERG fusion proteins in many cancer types [86,87,88], however, limited studies reported the role of ERG in the regulation of tumor neovascularization. For instance, Nagai et al. in a mouse xenograft B16F0 tumor model, which depends on angiogenesis for growth, observed that knocking out endothelial ERG significantly reduced the size of melanoma tumors, and significantly reduced tumor blood vessel density and pericyte coverage compared to controls [89]. This study, therefore, confirms that ERG could play an essential role in tumor angiogenesis and growth and that downregulation of ERG expression could be an effective strategy towards developing new anticancer therapies.

As ERG is largely involved in the biology of cancer, it can be considered as a potential new target for cancer therapies itself. Indeed, ERG is one of the most overexpressed oncogenes in prostate cancer, where a chromosomal translocation results in the fusion of the promoter region of androgen-regulated transmembrane protease serine 2 (TMPRSS2) with the DNA-binding domain of ERG [86]. TMPRSS2-ERG expression leads to the upregulation of the histone deacetylase 1 (HDAC) gene and the downregulation of its target genes. A therapy based on the inhibition of HDAC can be, therefore, effective against prostate cancer development. The HDAC inhibitors can indeed reduce cancer growth by inducing apoptosis of ERG positive prostate cancer cells [88]. Moreover, the development of YK-4-279, a small molecule inhibitor of ETS factors, reduced invasion, motility, and metastasis of ERG positive cells in prostate cancer [90].

The ERG DNA-binding activity can also be targeted by modulators, such as DB1255, which prevents ERG DNA binding [91]. Finally, the degradation of ERG by targeting an ubiquitin-specific peptidase 9, resulted in prostate tumor growth inhibition, both, in vitro and in vivo [92]. Altogether, these therapeutic approaches, developed at present as anti-tumor strategies, might also influence tumor angiogenesis in view of its upregulation in TEC.

## 3. Extracellular Vesicles

Extracellular vesicles (EVs) are bio-active particles delimited by a lipid bilayer, secreted by a wide variety of cells, nowadays emerging as one of the main effectors of intercellular communication [93]. Depending on the cell source, EVs have been shown to exert multiple effects on specific cell targets by modifying their function and phenotype [93].

The clinical use of EVs for the treatment of cancer is currently under evaluation, being these bioactive molecules an efficient tool to allow the delivery of therapeutic cargos to neoplastic cells [94]. In oncology, the use of EVs has been proposed not only as a biological carrier for anti-tumor drugs but also as an immunomodulator and tumor vaccination [95].

Among the possible EV sources, EVs isolated from stem cells are one of the most studied as an anticancer strategy [96]. However, few studies investigated their direct effects on tumor angiogenesis [97,98,99]. Lee et al. showed that EVs isolated from mesenchymal stem cells (MSC) were able to inhibit tumor growth and angiogenesis in a murine model of breast cancer, by downregulating VEGF production in breast cancer cells [98]. On the other hand, EVs isolated from cardiosphere-derived cells were able to inhibit tumor angiogenesis in a murine model of fibrosarcoma [99].

More recently, Lopatina et al. showed that EVs derived from human liver stem cells (HLSC) exhibit a direct anti-angiogenic effect on tumor-derived endothelial cells isolated from human renal carcinoma [97]. Treatment of renal tumor endothelial cells with HLSC-EVs in vitro inhibited the angiogenic and migration properties of TEC in a dose-dependent manner. The inhibitory effects on angiogenesis were mainly attributed to a down-regulation of different proangiogenic genes, targets of specific miRNAs enriched in HLSC-EVs. In addition, the anti-angiogenic activity of HLSC-EVs has been observed in vivo in a model of tumor angiogenesis in SCID mice. HLSC-EVs treated tumor endothelial cells showed a limited ability to connect with murine vasculature, and treatment of pre-existent tumor vessels with HLSC-EVs reduced vessel density [97].

These studies reveal a complex effect of EVs on tumor vascularization, which may result from the modulation of multiple targets on different tumor cell types, including a direct effect on TEC and an indirect one on tumor cells.

## 4. Anti-Angiogenic Vaccination

Cancer vaccines are emerging as one of the most promising tools for tumor eradication. Recently, new vaccination strategies against TEC, rather than against cancer cells, have been proposed [100]. Being TEC phenotypically and genotypically different from a normal endothelium [22], anti-angiogenic vaccination would theoretically target the activated tumor endothelium only, without affecting other angiogenic processes involving normal endothelial cells. On the other hand, targeting one specific molecule could activate compensatory angiogenic pathways and resistance mechanisms, that could be overcome by the combination with tumor immunotherapy or chemotherapy, or with other endothelial-cell vaccines.

Vaccination protocols involve the use of different vaccine types, such as DNA or peptide vaccines, or directly the injection of blocking antibodies against different immunogenic epitopes of proteins overexpressed by TEC [98,99,100,101,102]. However, both in preclinical and in clinical studies, the efficacy and the observed adverse events were variable, according to vaccine type, route of administration and to the adjuvant choice [98,99,100,101,102]. At present, different anti-angiogenic vaccination protocols, involving the use of peptide-based vaccines, are undergoing clinical trials, as summarized in Table 3 [101,102,103,104,105,106,107,108].

Preclinical studies involving the development of an immune response against different antigens overexpressed by TEC, such as bFGF, angiomotin, endoglin, Robo4, PDGFRβ, Tie-2, and tumor endothelial markers (TEM1 and TEM8) show that endothelial vaccination successfully reduces tumor growth in different tumor models, both in vitro and in vivo [100]. However, TEC genetic instability, together with the activation of compensatory pathways, may lead to an incomplete response to vaccination therapies against specific targets. Therefore, further clinical studies using whole endothelial [106] or placental cells [107] to induce a polyvalent immune response are currently under evaluation.

## 5. Vascular Normalization and Detransformation

Vessel normalization is defined as a vascular remodeling that leads to the re-acquisition of a normal structure and function of abnormal vessels [109].

The concept of vascular normalization as a therapeutic strategy to improve chemotherapeutic drug delivery to tumor cells was introduced in 1996 when Yuan et al. observed an increase of permeability in tumors treated with a VEGF-neutralizing antibody [110]. Several combinations of classical anti-angiogenic agents and cytotoxic drugs (used in a low dose and continuous protocol, the so-called metronomic dose) were, therefore, studied in clinical trials, but only a marginal increase of antitumor efficacy was observed [111]. Indeed, both the dose and the temporal window of anti-angiogenic treatment needed to achieve a transient normalized vasculature, which allows an adequate drug delivery to the inner tumor mass showed high variability [112,113].

Alternative strategies to achieve an increased response to anti-tumor therapies involving vascular normalization have been proposed during the past years. Class 3 semaphorins (Sema3) are secreted proteins that regulate cell adhesion through the signaling mediated by their receptors, composed by the dimerization of neuropilins and plexins [114]. In endothelial cells, neuropilins were found to bind VEGF receptors, regulating vascular development [114]. Sema3 acts as a tumor suppressor by blocking tumor cell growth and invasion, as well as by inducing endothelial cell apoptosis [114]. In addition, Sema3A has been identified as a novel normalizing agent that can overcome the resistance to anti-angiogenic therapies by extending the normalization window in mouse models [115].

Moreover, combined therapies involving activation of immune response and classical anti-angiogenic agents used at a normalizing dose are under study. Several groups recently demonstrated that the immunotherapy effect is enhanced by vessel normalization [109,113,116,117,118,119]. In particular, the immune checkpoint blockade of the programmed death receptor-1 (PD-1)/PD ligand 1 (PD-L1) pathway, combined with the VEGF-pathway blockade, can enhance both anti-tumor immunity and a structural normalization of tumor vessels [116,118,119]. In particular, Schmittnaegel et al. observed that blocking both angiopoietin and VEGF pathways induced tumor vessel normalization that favored a cytotoxic immune response [118]. Allen et al. contemporarily observed that combination therapy using blocking antibodies against VEGFR2 and PD-L1 resulted in enhanced cytotoxic activity, together with an increased normalizing effect of VEGF blockade on tumor vasculature [119]. Clinical trials investigating the efficacy of a combined therapy that involves the use of immune checkpoint inhibitors and anti-angiogenic agents are currently undergoing [120].

All the anti-angiogenic therapies mentioned above, including normalizing therapies, may induce overtime a transformation of the tumor-associated endothelial cells towards a mesenchymal profile, called endothelial to mesenchymal transition (EndMT) [121]. In the course of EndMT, resident endothelial cells delaminate from an organized cell layer and acquire a mesenchymal phenotype characterized by loss of cell–cell junctions, loss of endothelial markers, the gain of mesenchymal markers, and acquisition of invasive and migratory properties [120]. In cancer, EndMT supports the formation of cancer-associated fibroblasts, which are known to facilitate tumor progression. Furthermore, EndMT could also modify the endothelium abnormally, therefore, assisting tumor-cell extravasation. Lastly, EndMT has also been reported to be induced by events such as hypoxia, high glucose levels, as well as through the release of soluble factors in the tumor microenvironment [121].

Nagai et al. showed that EndMT can be triggered through a reduction in the expression of ERG together with friend leukemia integration 1 transcription factor (FLI1), which has been reported to play a pivotal role in endothelial cell homeostasis. A combined knockdown of both ERG and FLI-1 through short interfering RNA (siRNAs) in endothelial cells, caused the downregulation of endothelial genes accompanied by a consistent upregulation of genes involved in EndMT, such as alphaSMA and CollagenA1 in vitro [89].

It can, therefore, be concluded that dysregulation of angiogenic signaling pathways that play a crucial role in the homeostasis of endothelial cells, can cause an imbalance in endothelial physiology, leading to EndMT, which has been implicated in cancer progression.

As low doses of anti-angiogenic therapies from one side may induce the transformation of aberrant vases towards normal vessels, and from the other side may favor the activation of a mesenchymal phenotype in endothelial cells, an optimal therapy should take into consideration both vascular normalization and endothelial de-mesenchymalization.

For example, the combination of VEGF-targeting agents and TGFβ signaling inhibition, such as Sunitinib and TRC105 [51], as discussed above, could represent a valid therapy to block both endothelial and mesenchymal-related pathways.

## Figures and Tables

**Figure 1 ijms-20-06180-f001:**
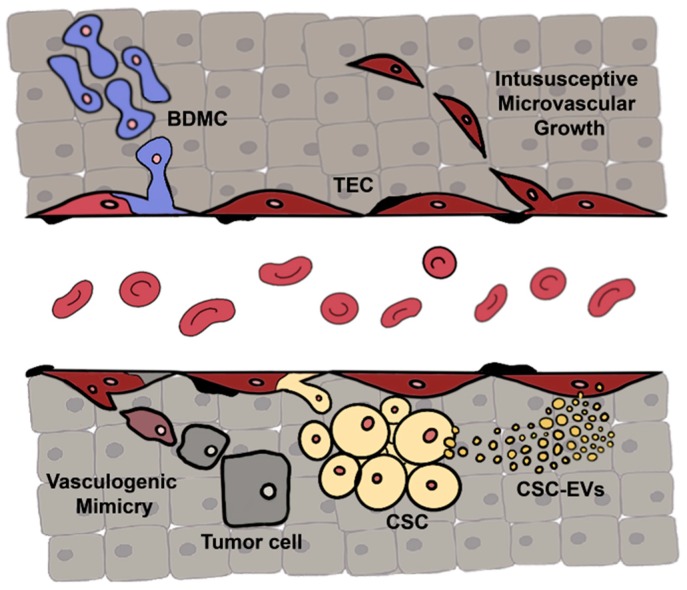
Alternative strategies of tumor vascularization. Tumor vessels may be generated by intra-tumor vasculogenesis as an alternative to endothelial cell recruitment from adjacent vessels. TEC may originate from the recruitment of bone marrow-derived cells (BMDC), such as endothelial progenitor cells, or directly from tumor cells acquiring an endothelial phenotype in a process called vasculogenic mimicry. Moreover, a subpopulation of cancer cells with stem features (CSCs) can directly differentiate into tumor endothelial cells (TEC) or can reprogram normal endothelial cells by the release of extracellular vesicles (CSC-EVs). Finally, intussusceptive microvascular growth allows the generation of a new vessel by the split of a pre-existing one.

**Figure 2 ijms-20-06180-f002:**
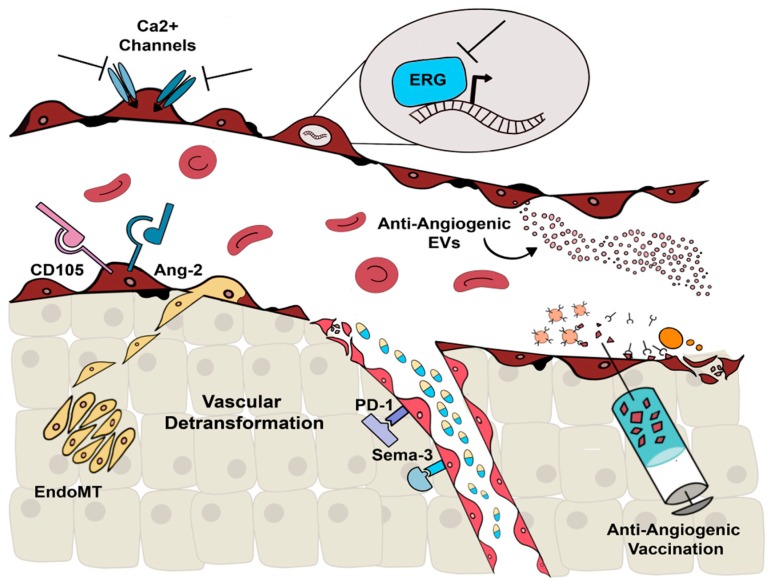
Alternative strategies to target tumor vascularization. Approaches to overcome the resistance to classical anti-angiogenic agents may involve the target of different molecules, such as calcium-permeable channels (Ca^2+^ channels), the transcription factor ERG, endoglin (CD105), or angiopoietin (Ang-2). TEC could also be targeted by stem cell-derived extracellular vesicles with anti-angiogenic effect (anti-angiogenic EVs), or by a specific multi-targeted cytotoxic immune response driven by anti-angiogenic vaccination. The irregular vascular network could be targeted by new normalizing agents, such as Sema 3. Finally, endothelial–mesenchymal transition (EndoMT), involving the downregulation of angiogenic molecules, represents an additional strategy for anti-angiogenic therapy resistance. Vascular detransformation represents, therefore, a novel strategy to block tumor abnormal vascularization.

**Table 1 ijms-20-06180-t001:** TEC isolation from solid tumors.

Tumor Type	Species	Year	References
Glioblastoma	Human	1999	Alessandri et al. [23]
Colon	Human	2000	St. Croix et al. [5]
Brain tumors	Human	2002	Unger et al. [24]
Renal	Human	2003	Bussolati et al. [7]
Lung	Mouse	2003	Allport et al. [25]
B-Cell lymphoma	Human	2004	Streubel et al. [26]
Liposarcoma and melanoma	Mouse	2004	Hida et al. [27]
Breast	Human	2006	Grange et al. [28]
Breast	Mouse	2006	Amin et al. [29]
Liver	Human	2007	Wu et al. [30]
Ovary	Human	2007	Buckanovitch et al. [31] Lu et al. [32]
Glossal lymphangioma	Human	2010	You et al. [33]
Prostate	Human	2014	Fiorio et al. [8]

**Table 2 ijms-20-06180-t002:** Main anti-angiogenic drugs for solid tumors treatment.

Drug Name	Type	Targets	Tumor Type	Combined Therapy
Bevacizumab	mAb	VEGF-A	Colorectal, lung, glioblastoma, renal cell carcinoma, breast, brain, ovarian, cervical, fallopian tube, and peritoneal cancer	Fluoropirimidine, Cisplatinum, Paclitaxel, Interferon a-2a
Sorafenib	TKI	VEGFR1/2/3, PDGFR, c-kit	Renal cell carcinoma, liver, thyroid, desmoid tumors	
Sunitinib	TKI	VEGFR1/2/3, PDGFR, c-kit, FLT-3, Ret	Renal cell carcinoma, gastrointestinal stromal, pancreatic neuroendocrine cancer, and leukemia	
Pazopanib	TKI	VEGFR1/2/3, PDGFR, c-kit, FGFR	Renal cell carcinoma and soft tissue sarcoma	
Axitinib	TKI	VEGFR1/2/3, c-kit, PDGFR	Renal cell carcinoma	
Regorafenib	TKI	VEGFR1/2/3, PDGFRα/β, FGFR1/2, Tie2, c-Kit	Metastatic colorectal cancer, advanced gastrointestinal stromal cancer and advanced hepatocellular carcinoma	
Cabozantinib	TKI	c-MET, VEGFR2, AXL, Ret	Medullary thyroid cancer and renal cell carcinoma	
Nintedanib	TKI	VEGFR1/2/3, PDGFR, FLT-3	Idiopatic pulmonary fibrosis, lung cancer	Docetaxel
Levantinib	TKI	VEGFR1/2/3, PDGFR, FGFR, Ret, c-Kit	Thyroid cancer and renal cell carcinoma	Everolimus
Vandetanib	TKI	VEGFR1/2/3, EGFR, and Ret	Medullary thyroid cancer	

**Table 3 ijms-20-06180-t003:** Main anti-angiogenic vaccination approaches currently undergoing clinical trials.

Antigens	Vaccine Type	Tumor Type	Phase	REF/NIH N.
VEGF-A	Recombinant human VEGF-A-121 isoform	Advanced solid tumors	I	Gavilondo 2014 [108]
VEGFRs	VEGFR2-169 peptide	Pancreatic cancer	I	Miyazawa 2010 [101]
	VEGFR1-1084 and VEGFR2-169 peptides		I/II	NCT00655785
	VEGFR1-A2-770 peptide		I/II	NCT00683085
	VEGFR2-169 peptide	Advanced solid tumors	I	Okamoto 2012 [102]
	VEGFR1-1084 peptide		I	Hayashi 2013 [103]
	VEGFR2, VEGFR1, URLC10, TTK, CDCA1 multipeptide	Non small cell lung cancer	I	Suzuki 2013 [104]
Survivin	hTERT/survivin/CMV multipeptide	Breast cancer	I	NCT01660529
	Survivin long peptide	Neuroendocrine tumors	I	NCT03879694
	Salmonella-based Survivin peptide	Multiple myeloma	I/II	NCT03762291
EGF	Recombinant Human EGF-rP64K/Montanide ISA 51 peptide	Non-small cell lung cancer	II	Garcia 2008 [105]
			II/III	NCT00516685
			III	NCT02187367
			III	NCT01444118
		Non-small cell lung cancer, squamous head and neck cancer	I/II	NCT02955290

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
