# Peer review of "Alternative Strategies to Inhibit Tumor Vascularization"

_ijms, 2019, doi:10.3390/ijms20246180_

Round 1
Reviewer 1 Report
The manuscript is devoted to the analysis of experimental data in a very significant field. The authors critically described the currently available experimental data on the possibility of treating tumors by reducing their vascularization.
I would like to see (in the future works of the authors) more experimental details confirming the concept of endothelial demesenchymalization.
Only minor corrections to text editing should be done (line 241 "form" -"from", lines 286, 307, and some others "at al" - " at al.", line 288 "in low" - "in low dose", line 323 - "have both been" ?, lines 351, 393, 514 require clarification.
Author Response
We thank the reviewer for the careful evaluation and for pointing out some typos. We modified the text according to the reviewer’s suggestions. In particular line 241 "form" has been changed to "from", lines 286, 307, and some others "at al" " has been changed to " et al.", line 288 "in low" - " has been changed to "in low dose", line 323 - "have both been", has been changed to have been.
Reviewer 2 Report
In this review, Brossa et al. explain the role of tumor endothelial cells (TEC) in the neovascularization that occurs during the “angiogenic switch” in cancer progression. Authors describe the classic anti-angiogenic therapies as well as alternative strategies to inhibit tumor vascularization.
Some points should be revised:
Recent studies describe the role the MP0250 neutralizing DARPin® molecule as new anti-angiogenic strategy (Binz HK et al, Mabs 2017; Fiedler U et al, Oncotarget 2017, Rao L et al, Oncotarget 2018). The potential anti-tumor and anti-angiogenic role of this molecule should be discussed. The role of combination therapy based on the simultaneous inhibition of angiogenesis and immunomodulatory pathway (Allen E et al, Sci Transl Med. 2017; Yi M et al, Molecular Cancer 2019) should be included.Author Response
Recent studies describe the role the MP0250 neutralizing DARPin® molecule as new anti-angiogenic strategy (Binz HK et al, Mabs 2017; Fiedler U et al, Oncotarget 2017, Rao L et al, Oncotarget 2018). The potential anti-tumor and anti-angiogenic role of this molecule should be discussed.
Response: We thank the reviewer for the suggestion, that we followed by adding a paragraph describing the role of MP0250 in the “Alternative anti-angiogenic antibodies” chapter (lines 135-141).
The role of combination therapy based on the simultaneous inhibition of angiogenesis and immunomodulatory pathway (Allen E et al, Sci Transl Med. 2017; Yi M et al, Molecular Cancer 2019) should be included.
Response: We added and discussed the reference Yi M et al Molecular Cancer 2019 in line 327, and the study of Allen E et al in lines 323-326.
Round 2
Reviewer 1 Report
no
Reviewer 2 Report
In this review, Brossa et al. explain the role of tumor endothelial cells (TEC) in the neovascularization that occurs during the “angiogenic switch” in cancer progression. Authors describe the classic anti-angiogenic therapies as well as alternative strategies to inhibit tumor vascularization.
Revisions improved the scientific relevance of the paper.
Authors have covered all my questions. I have no more queries.